# Whole-Genome Sequencing Revealed the Fusion Plasmids Capable of Transmission and Acquisition of Both Antimicrobial Resistance and Hypervirulence Determinants in Multidrug-Resistant *Klebsiella pneumoniae* Isolates

**DOI:** 10.3390/microorganisms11051314

**Published:** 2023-05-17

**Authors:** Andrey Shelenkov, Yulia Mikhaylova, Shushanik Voskanyan, Anna Egorova, Vasiliy Akimkin

**Affiliations:** Central Research Institute of Epidemiology, Novogireevskaya Str., 3a, 111123 Moscow, Russia

**Keywords:** *Klebsiella pneumoniae*, whole genome sequencing, antimicrobial resistance, fusion plasmids, hypervirulence, incompatibility groups, MDR

## Abstract

*Klebsiella pneumoniae*, a member of the *Enterobacteriaceae* family, has become a dangerous pathogen accountable for a large fraction of the various infectious diseases in both clinical and community settings. In general, the *K. pneumoniae* population has been divided into the so-called classical (cKp) and hypervirulent (hvKp) lineages. The former, usually developing in hospitals, can rapidly acquire resistance to a wide spectrum of antimicrobial drugs, while the latter is associated with more aggressive but less resistant infections, mostly in healthy humans. However, a growing number of reports in the last decade have confirmed the convergence of these two distinct lineages into superpathogen clones possessing the properties of both, and thus imposing a significant threat to public health worldwide. This process is associated with horizontal gene transfer, in which plasmid conjugation plays a very important role. Therefore, the investigation of plasmid structures and the ways plasmids spread within and between bacterial species will provide benefits in developing prevention measures against these powerful pathogens. In this work, we investigated clinical multidrug-resistant *K. pneumoniae* isolates using long- and short-read whole-genome sequencing, which allowed us to reveal fusion IncHI1B/IncFIB plasmids in ST512 isolates capable of simultaneously carrying hypervirulence (*iucABCD*, *iutA*, *_p_rmpA*, *peg-344*) and resistance determinants (*armA*, *bla_NDM-1_* and others), and to obtain insights into their formation and transmission mechanisms. Comprehensive phenotypic, genotypic and phylogenetic analysis of the isolates, as well as of their plasmid repertoire, was performed. The data obtained will facilitate epidemiological surveillance of high-risk *K. pneumoniae* clones and the development of prevention strategies against them.

## 1. Introduction

*Klebsiella pneumoniae* is a significant hospital pathogen that can cause infections of varying severity in the bloodstream, respiratory and nervous systems, and urogenital tract of a human or animal body, often leading to serious adverse events [1]. This Gram-negative bacterium is one of the most widespread and dangerous members of the Enterobacteriaceae family, and it has become one of the major causes of healthcare-associated infections worldwide [2].

Two independent evolutionary branches (pathotypes) have been considered to represent the *K. pneumoniae* population: classical (cKp) and hypervirulent Klebsiella (hvKp) [3,4]. CKp is distributed globally and constitutes a fraction of the opportunistic flora of healthy people. Such strains can rapidly acquire multiple resistance to antibiotics in clinical settings, which makes them a leading causative agent of nosocomial infections. A subgroup of cKp is designated as “high-risk clones” representing distinct genetic lineages that usually are multidrug-resistant (MDR) or extensively drug-resistant (XDR). Such isolates have spread worldwide and have caused large numbers of infections, mostly in clinical settings [5,6]. The best-known and most-characterized *K. pneumoniae* high-risk clones belong to multilocus sequence typing (MLST)-based types ST11, ST14, ST15, ST17, ST45, ST147, ST258, ST307 and ST512 [3].

HvKp is associated with more severe and aggressive infections, usually in healthy and community-residing individuals [7,8]. HvKp, more often than cKp strains, carries genetic factors associated with a hypermucoid phenotype and with the synthesis of lipo- and polysaccharide capsules, as well as genes encoding the elements of an iron acquisition system, allantoin utilization system, and type I fimbriae [8,9,10]. Such strains are predominantly associated with clones ST23, ST65 and ST86 [11].

CKp and hvKp were previously considered as non-overlapping evolutionary branches due to their significant differences [12]. However, during the last fifteen years, several research studies have demonstrated the acquisition of multidrug resistance by hypervirulent strains of Klebsiella in South Korea [13], Argentina [14], France [15], China [16] and the Russian Federation [17,18]. Moreover, a case of the acquisition of a virulence plasmid by a classical extremely resistant hospital strain of *K. pneumoniae* in China has been described [19]. This confirmed the concerns of some researchers regarding the possibility of a new “superpathogen” formation combining the pathogenetic properties of the two *K. pneumoniae* evolutionary branches [20].

Hypervirulence, as well as multidrug resistance, is usually associated with the acquisition of additional genetic material and the formation of genetic lines that effectively support these acquired determinants [21]. The ecology of *K. pneumoniae* and its genome plasticity designated this species as a key reservoir and distributor of antibiotic resistance genes. This pathogen is more competent to acquire genetic material of plasmid origin from different donors of different ecological niches [22]. This bacterium is also capable of keeping such plasmids long enough to pass them on to new recipients living in humans and animals [5]. *K. pneumoniae* often contains more than one plasmid in the genome, including both low-molecular-weight, high-copy-number, and high-molecular-weight, low-copy-number, plasmids. High-molecular-weight plasmids belong to different incompatibility groups and usually carry genetic determinants of virulence and antibiotic resistance [23]. In recent years, whole-genome sequencing (WGS), especially long-read sequencing, has become a powerful tool for the determination of plasmid structures, as well as antimicrobial resistance (AMR) and virulence gene locations [17,24,25].

In this study, we described the detailed phenotypic and genomic characteristics of four MDR *K. pneumoniae* isolates based on antimicrobial susceptibility testing, hybrid whole-genome assembly and phylogenetic comparison. The analysis performed allowed us to assess thoroughly the plasmid composition of the samples studied and to determine the localization of AMR and virulence determinants. We also succeeded in revealing the convergence of hvKP and MDR-cKp traits in one of the isolates, which resulted in the emergence of a strain that could pose a serious threat to the healthcare system. The data obtained will be useful for epidemiological surveillance of this important pathogen, especially in the Russian Federation, where it has become the major source of nosocomial infections in recent years (AMRmap database [26], https://amrmap.ru/, accessed on 1 March 2023).

## 2. Materials and Methods

### 2.1. Sample Collection, Susceptibility Testing and DNA Isolation

Three *K. pneumoniae* isolates were collected from different patients of the same medical department (ICU) during the short intervals (3–4 days) of their hospital stays. One *K. pneumoniae* sample was isolated from a patient in the pulmonology department during the same period. This patient was discharged from the hospital and the other three patients died. More detailed metadata are presented in Table 1.

The total number of bacterial isolates analyzed in the tertiary care hospital per year in the pilot study was approximately 850, of which 160 were sequenced and about 80% represented *K. pneumoniae*. The reasons for choosing these particular isolates for investigation were the severity of infections they caused, their resistance to several classes of antibiotics including carbapenems and their possible carriage of hybrid plasmid replicons revealed by PCR.

Species identification was performed using time-of-flight mass spectrometry (MALDI-TOF MS) with the VITEK MS (bioMerieux, Marcy-l’Étoile, France). Antimicrobial susceptibility/resistance of the isolates was determined by the disc diffusion method using the Mueller–Hinton medium (bioMerieux, Marcy-l’Étoile, France) and disks with antibiotics (BioRad, Marnes-la-Coquette, France), and by the Minimum Inhibitory Concentration (MIC) method using a VITEK 2 Compact 30 analyzer (bioMerieux, Marcy-l’Étoile, France).

The antibiotics tested included amikacin, amoxicillin, ampicillin, cefepime, cefotaxime, ceftazidime, ceftriaxone, ceftazidime/avibactam, cefuroxime, ciprofloxacin, ertapenem, gentamicin, imipenem, levofloxacin, meropenem, piperacillin/tazobactam, tobramycin and trimethoprim/sulfamethoxazole. EUCAST clinical breakpoints, version 11.0 (https://www.eucast.org/clinical_breakpoints/, accessed on 20 December 2021), were applied to interpret the susceptibility/resistance results obtained.

### 2.2. Whole-Genome Sequencing

Genomic DNA was isolated with the DNeasy Blood and Tissue kit (Qiagen, Hilden, Germany); then, the Nextera™ DNA Sample Prep Kit (Illumina^®^, San Diego, CA, USA) was applied to prepare paired-end libraries for WGS of the isolates on an Illumina^®^ NextSeq 2000 platform (Illumina^®^, San Diego, CA, USA).

The same extracted genomic DNA was used for library preparation for the Oxford Nanopore MinION sequencing system (Oxford Nanopore Technologies, Oxford, UK) with the Rapid Barcoding Sequencing kit SQK-RBK004 (Oxford Nanopore Technologies, Oxford, UK). The amount of initial DNA was 400 ng for each sample. The libraries were prepared according to the manufacturer protocols and were sequenced on FLO-MIN106 R9.4 flow cell with a standard 24 h sequencing protocol using the MinKNOW software version 22.03 (Oxford Nanopore Technologies, Oxford, UK).

### 2.3. Genome Assembly, Data Processing and Annotation

Basecalling of the raw MinION data was made using Guppy basecalling software version 6.1.7 (Oxford Nanopore Technologies, Oxford, UK), and demultiplexing of the samples was performed using Guppy barcoding software version 6.1.7 (Oxford Nanopore Technologies, Oxford, UK).

Hybrid short- and long-read assemblies were obtained using Unicycler version 0.5.0 (normal and bold mode) [27]. Additional plasmid assembly was performed using Flye software, version 2.9.1 [28], using default parameters, and assembly polishing and correction was made with Polypolish version 0.5.0 [29] and Medaka version 1.7.3 (https://github.com/nanoporetech/medaka, accessed on 1 March 2023) using the ‘r941_min_fast_g507’ model.

Genome assemblies were submitted to NCBI Genbank under the project PRJNA942929.

The pipeline described earlier [30,31] was used for the raw data filtration and assembled genome processing and annotation. Resfinder 4.3.0 software was used for antimicrobial gene detection (https://cge.cbs.dtu.dk/services/ResFinder/, accessed on 20 February 2023, using default parameters) and point mutation analysis. Virulence factors were searched in VFDB (http://www.mgc.ac.cn/VFs/main.htm, accessed on 20 February 2023, using default parameters). Additional virulence genes (*peg-344*, etc.) were searched using BLASTn using reference sequences from Genbank (https://www.ncbi.nlm.nih.gov/genbank/, accessed on 12 March 2023).

Plasmid replicon types, the predicted mobility and other annotation features were detected using MOB-suite [32] using default parameters (mob_typer version 3.1.2). Plasmid visualization was made using BRIG (version 0.95, https://github.com/happykhan/BRIG, accessed on 12 March 2023). CRISPRCasFinder [33] was used to identify the presence of CRISPR/Cas systems in the genomes studied. Spacers were analyzed using the CRISPRTarget tool [34].

Isolate typing was performed by MLST using BIGSdb (https://bigsdb.pasteur.fr/klebsiella/, accessed on 20 February 2023). In addition, the types based on capsule synthesis loci (K-loci) and lipooligosaccharide outer core loci (OCL) were detected using Kaptive software version 2.0.3 with default parameters [35]. Virulence scores for the isolates were assessed with Kleborate [36] version 2.3.2.

Detection of cgMLST profiles was performed using MentaList (https://github.com/WGS-TB/MentaLiST, version 0.2.4, default parameters) using the scheme obtained from cgmlst.org (https://www.cgmlst.org/ncs/schema/schema/2187931/, contained 2358 loci, last update 20 February 2023) [37]. The minimum spanning tree was built using PHYLOViz online (http://online.phyloviz.net, accessed on 20 February 2023).

## 3. Results

### 3.1. Isolate Typing and Resistance Profiles

In silico MLST revealed that three isolates (CriePt492, 494 and 495) belonged to ST512 and they were characterized as possessing KL107 capsular type and O2v2 oligosaccharide type. The typing results showed another profile for the isolate CriePt491—ST147/KL20/O2v1. It is worth noting that the determined genetic lines belong to the “high-risk clones” of *K. pneumoniae* mentioned above.

According to phenotypic analysis, all isolates were susceptible to ceftazidime/avibactam and resistant to the other antimicrobial compounds tested. The isolates CriePt492 and CriePt495 were additionally susceptible to gentamycin. Thus, all the isolates under investigation exhibited multidrug resistance (Figure 1).

CgMLST analysis revealed that the genomic sequences of ST512 isolates were very similar (3 allele differences between CriePt492 and 495, and 29 different alleles between either of these isolates and CriePt494). Thus, according to the criterion described previously (≤18 cgMLST allele differences between the isolates to be considered as belonging to a single strain or clone for *K. pneumoniae* [38]), the isolates CriePt492 and 495 were highly likely representing a single strain; CriePt494 demonstrated the beginning steps of divergence from its relative samples.

In silico searching for antimicrobial resistance determinants showed that chromosomal genes (*oqxAB* and *fosA*) were represented in all the isolates under investigation. In addition, *fosA* and *bla_SHV-182_* were found on the chromosomes of CriePt492 and 495, and only *fosA* (without *blaSHV*, which was found on a plasmid) on the chromosome of CriePt494. In contrast, CriePt491 had the major part of its AMR genes on its chromosome.

Eleven antibiotic-resistance genes (ARGs) were found in both ST512 isolates, CriePt492 and 495, and their resistance gene profiles were completely the same. These isolates carried two aminoglycoside resistance genes, two beta-lactamase genes (*bla_KPC-3_* and *bla_SHV-182_*) and other genes associated with resistance to chloramphenicol (*cat*), macrolides (*mph(A)*), fosfomycin (*fos*), fluoroquinolones (*oqx*), trimethoprim (*dfr*) and sulfonamides (*sul*). Another ST512 isolate, CriePt494, differed from its relatives by having additional determinants, namely, two additional aminoglycoside resistance genes, two beta-lactamase genes (*bla_CTX-M15_* and *bla_NDM-1_*), as well as one more each of macrolide-, fluoroquinolone- and trimethoprim-resistance genes. The composition of the antimicrobial-resistance genes of the ST147 isolate (CriePt491) was more similar to the isolate CriePt494, and it carried one additional beta-lactamase gene, *bla_TEM-1A_*, thus possessing four different beta-lactam resistance genes in the genome. It also had one more gene associated with resistance to macrolides—*msr (E)*.

The differences also included the *armA* gene, which was revealed only in CriePt491 (chromosome) and CriePt494 (plasmid), and which corresponded to their phenotypic resistance to gentamicin.

Point mutation analysis revealed porin mutations associated with carbapenem resistance in all isolates. OmpK36 mutation A217S and ompK37 mutations I70M and I128M were revealed in all isolates, and ompK37 N230G was additionally found in ST512 isolates. Several mutations conferring resistance to fluoroquinolones were also revealed in the *acrR* gene for all isolates studied. A summary of point mutations revealed is provided in Appendix A.

Annotation results of the resistance determinants for the isolates studied were consistent with the resistance phenotypes for the antimicrobial drugs included in the panel.

### 3.2. Isolate Typing and Resistance Profiles

The sets of virulence factors in *K. pneumoniae* isolates under investigation were very similar and included up to 68 virulence genes. The most important of them, which were revealed in all four isolates, comprised acridine efflux pump genes (*acr*), fimbria involved in the processes of adhesion, formation of biofilms, other regulators of the initial stages of infection (*fimABCDEFGHIK*, *mrkABCDFHIJ*, *ecpRABCDE*, *RscAB*), genes encoding outer membrane, ATP-binding proteins and regions associated with type IV secretion system and enzymes of catalytic activity (*dotU*, *ompA*, *clpV*, *tssF/G*, *ugd*, *gnd*, etc.), as well as several siderophores (*entABCEF*, *fepABCDG*) and genes of iron uptake (salmochelin and aerobactin) clusters (*iutA* and *iroE*, respectively).

Two isolates (CriePt492 and 495) belonging to the single strain had exactly the same set of virulence factors. The differences were observed in the genome of the isolate CriePt494. It possessed the aerobactin cluster (*iucABCD*) and its receptor (*iutA*), and the capsule upregulation gene *_p_rmpA2* (where subscript *p* indicates its plasmid origin), a regulator of a mucoid phenotype. The plasmid CriePt494_p2 also contained *peg-344* metabolite transporter, which is considered one of the markers of hypervirulent phenotype [9]. The isolate CriePt491 differed by the presence of the *rfbK1* gene encoding O9 family phosphomannomutase involved in the synthesis of capsular polysaccharide.

The presence/absence for a selected set of the most important virulence genes for the isolates studied is shown in Figure 2. A complete list of virulence factors is given in Appendix A.

### 3.3. Plasmid Typing, Classification, Annotation and Comparison

The isolates studied differed from each other by plasmid number and composition. A brief description of replicon typing is given in Table 2, while the extended information for each plasmid assembled is shown in Table 3. Hybrid assembly allowed us to determine the refined plasmid structure and obtain the data regarding AMR and virulence gene location, which represents an important step in elucidating the possible mechanisms of their transfer.

CriePt492 and CriePt495 both included six plasmids with the same repertoire of replicons, while CriePt494 possessed an additional IncFIA plasmid. CriePt491 had the lowest number of plasmids (five) and did not possess the ColRNAI replicon. However, we revealed more differences in plasmid structure with a more detailed analysis. CriePr492, 494 and 495 shared the four smallest plasmids with lengths ranging from 2963 to 53,292, and their corresponding plasmids had the same or very close sequences. At the same time, the plasmids of CriePt491 were completely different despite having similar replicon types. Notably, the former three isolates had a small non-typeable plasmid with a length of 2963, while CriePt491 included a different, yet also non-typeable, plasmid with a length of 53,009.

Next, the plasmids of CriePt492, 494 and 495 will be discussed together since they shared many features, and the plasmids of CriePt491 will then be discussed separately.

#### 3.3.1. Plasmids of CriePt492, 494 and 495

The smallest non-typeable plasmid (length = 2963) had a 99% similarity with the pR17.4849_3.0k plasmid from *Salmonella enterica* obtained in Taiwan (Genbank acc. CP100750), which also did not have a commonly known replicon type and was attributed to the AG463 cluster by the MOB-typer tool. This plasmid was also sequenced on MinION, and sequence mismatches probably reflected the differences in the assembly algorithms used. The relaxases of MOB_V_ type were detected in these plasmids, and they were mobilizable, which means that they rely on conjugative plasmids to provide the mating pair formation components, and MPF_T_ plasmids can serve as helpers for their conjugation [39]. No AMR or virulence genes were found in these plasmids.

The IncX3 plasmids contained the gene of class A beta-lactamase *bla_KPC-3_* known to provide resistance to various antibiotics including third-generation cephalosporins and carbapenems for *K. pneumoniae*, as well as the *bla_SHV-182_* beta-lactamase gene, which was also revealed in the chromosomes. The plasmids were found to be conjugative and included MOB_P_ relaxase. In addition, they were attributed to MPF_T_ type, which is based on the plasmids’ T4SS system, involved in mating pair formation (MPF) during conjugation [40]. The closest Genbank match for this plasmid was the *Escherichia coli* strain SUISSEKPC3NDM5 plasmid p1606b (Genbank acc. CP083703.1, Switzerland), which shared higher than 99% identity with our plasmids, contained the same AMR genes and had the same total length.

The plasmids with the Col(pHAD28) replicon carried the single AMR gene *aac(6′)-Ib*, providing resistance to aminoglycosides. Col(pHAD28) plasmids had almost identical sequences in ST512 isolates and carried the *aac(6′)-Ib* gene providing aminoglycoside resistance. No relaxases were detected, and the plasmids were considered as non-mobilizable. The closest match from Genbank for these plasmids was pCR14_5 from *K. pneumoniae* (Genbank acc. CP015397.1, USA), which had a length of 9456 and included the same AMR gene.

The ColRNAI plasmids also had nearly identical sequences for the isolates studied. AMR genes and virulence factors were not revealed in them. The plasmids included MOB_C_ relaxase and were considered mobilizable. The *K. pneumoniae* strain BA4656 plasmid pColRNAI (Genbank acc. CP035911.1, length = 9729, India) had a higher than 99% similarity with this set of plasmids. The reference plasmid also did not include any virulence or resistance determinants.

CriePt494 possessed an additional conjugative IncFIA plasmid with a relaxase of MOB_F_ type. It carried important AMR determinants, such as the gene encoding class A extended-spectrum beta-lactamase (ESBL), *bla_CTX-M-15_*, which can provide resistance to various beta-lactam antibiotics including fourth-generation cephalosporins. This gene was not found in other ST512 isolates, and was revealed in CriePt491, but in a plasmid of different type. Additional AMR genes located on this IncFIA plasmid included *erm(B)*, *aac(3)-IIa* and *qnrS1*, providing resistance to lincosamides/macrolides, aminoglycosides and fluoroquinolones, respectively. A similarity among reference Genbank plasmids was found in the *Shigella sonnei* plasmid p6904-27 (CP045525.2, Switzerland, length = 83,273) and *Escherichia coli* strain SUISSEKPC3NDM5 plasmid p1606d (CP083705.1, Switzerland, length = 146,558), which also included the *bla_CTX-M-15_* gene.

The IncFIB plasmids of CriePt 492 and 495 were very similar and included MOB_F_ relaxase. However, they differed by one AMR gene—*aac(6′)-Ib*—which was revealed in a plasmid from CriePt495, but not in CriePt492. Other AMR genes were the same for both plasmids and included *aadA2*, *aph(3′)-Ia*, *catA1*, *dfrA12*, *mph(A)* and *sul1,* thus empowering these isolates with a broad range of weapons against various classes of antimicrobial drugs. In contrast, the IncFIB plasmid of CriePt494 was almost twice as short and did not include any AMR genes, while most of the genes given above were revealed in the IncHI1B plasmid of this isolate. In addition, the CriePt494 plasmid did not possess any relaxases. CriePt494_p3 was similar to the pIT-394-FIB *K. pneumoniae* plasmid (CP110954.1, Italy), which had almost the same length and did not include any AMR genes either. CriePt492_p3 was similar to the pKPN-a68 plasmid (CP009777.1, USA, length = 212,192), which included the same AMR genes.

Meanwhile, the main differences inside the group of ST512 isolates (CriePt 492, 494 and 495) were revealed in the large fusion conjugative plasmids IncHI1B/IncFIB (p2). All three plasmids were attributed to MOB_H_ mobility type and to MPF_F_ type. However, while the plasmids from CriePt492 and 495 were almost completely similar and did not carry any AMR genes or virulence factors, the differences between CriePt494_p2 and its counterparts were dramatic. This plasmid included both AMR genes, including the New Delhi metallo-beta-lactamase-1 gene and other genes providing resistance to several classes of antibiotics, and virulence factors *iucABCD* and *rmpA2*. As was mentioned above, CriePt494 was the only isolate carrying these hypervirulence factors. In addition, it also included *peg-344.* This plasmid was similar to a fusion plasmid, phvKpST147_NDM-1_2566 (MW911671.1), revealed recently in hypervirulent *K. pneumoniae* from Russia, but it had a slightly different repertoire of virulence/resistance genes. All other virulence genes revealed in the isolates studied were found in the chromosomes.

A circular diagram representing the IncHI1B/IncFIB plasmids from CriePt isolates and the phvKpST147_NDM-1_2566 reference plasmid is shown in Figure 3. It is easily seen that the plasmids of CriePt 491, 492 and 495 lack the parts containing resistance and virulence genes, and phvKpST147_NDM-1_2566 lacks the regions containing *aph(3′)-Ia*, *dfrA12* and *aadA2* genes from CriePt494.

#### 3.3.2. Plasmids of CriePt491

Although CriePt491 was also MDR, as were its counterparts, a large fraction of the AMR genes in this isolate were located in its chromosome, while additional resistance determinants were revealed in the IncR plasmid not possessed by ST512 isolates. The plasmids of CriePt491 were mostly non-mobilizable and did not include any relaxases.

The large IncHI1B/IncFIB plasmid did not include any AMR genes or virulence factors. It had a region of more than 100,000 in length almost identical to the phvKpST147_NDM-1_2566 plasmid of *K. pneumoniae* (MW911671.1, Saint Petersburg, Russia), but multiple AMR genes and virulence factors in this 350k reference plasmid were located outside of this region.

The 100k IncFIB plasmid did not include any pathogenicity determinants either, but it possessed the gene encoding the RepB replication initiation protein. The closest match for this plasmid in Genbank was a phvKpST147_3 plasmid (CP066857.1, Saint Petersburg, Russia) with 99.9% identity, which also did not possess any AMR or virulence determinants.

The 53k p4 was similar to a plasmid from a Norwegian *K. pneumoniae* isolate of approximately the same length (CP034049.1), for which no known replicons were revealed either.

The small 5k Col(pHAD28) plasmid was similar to a P7 plasmid from a Spanish *K. pneumoniae* isolate (OW970547.1).

Finally, the plasmid of particular interest was a 47k IncR plasmid, which carried multiple AMR genes. The plasmid included three beta-lactamases, namely, class A *bla_CTX-M-15_* and ESBL *bla_TEM-1B_*, as well as class D *bla_OXA-9_*, which provided resistance to multiple beta-lactam antibiotics, including fourth-generation cephalosporins. In addition, aminoglycoside- and ciprofloxacin-resistance genes were detected. Together with the chromosome-encoded carbapenemase *bla_NDM-1_* and other resistance genes, these plasmids provided CriePt491 with a multifunctional antibiotic defense system. However, the IncR plasmid did not include any relaxases or mobilization proteins, and thus it requires additional helper plasmids to be transferred via a conjugation process. This plasmid was similar to the EFN 299 plasmid p4 (CP092593.1, Ghana) and phvKpST147_4 (CP066858.1, Saint Petersburg, Russia), which also carried multiple AMR genes including beta-lactamase genes.

### 3.4. CRISPR-Cas Systems and Anti-CRISPR Genes

A type I-E CRISPR-Cas system was identified in the CriePt491 chromosome (start = 896,591; end = 905,034) that included intact and apparently functional *cas2*, *cas1*, *cas6*, *cas5*, *cas7*, *cse2*, *cse1* and *cas3* genes. No reliable CRISPR sequences were identified in other isolates. The scheme of the system is shown in Figure 4.

Spacer analysis for the CriePt491 CRISPR-Cas system revealed a large fraction (about 70%) of anti-plasmid sequences, in particular, to IncFII and IncX3 plasmids. However, detailed analysis of these sequences lies beyond the scope of the manuscript.

Remarkably, the genes encoding the AcrIIA7 Cas9 inhibitor protein were found with >90% amino acid sequence identity in the chromosomes of all isolates except CriePt491.

### 3.5. Phylogenetic Analysis of the Isolates

We compared the ST512 isolates and ST147 CriePt491 separately with the corresponding reference isolates from Genbank (https://www.ncbi.nlm.nih.gov/genbank/, accessed on 10 February 2023) based on cgMLST analysis. The minimum spanning tree for the ST512 isolates and the best matches from Genbank are shown in Figure 5.

Pairwise comparison of cgMLST profiles according to the threshold set previously (≤18, [38]) revealed two isolates that were within the single strain range with CriePt492 and CriePt495—strain 52/1 (GCA_022555555 from Samara, Russia, 17 allele differences), and 2021CK-01402 (GCA_021904695.1, USA, 11 allele differences), while the closest neighbor of CriePt494 (again, GCA_021904695.1) had 26 allele differences, thus exceeding the threshold value. Both reference isolates contained various AMR genes, including *bla_KPC-3_*, but did not possess the hypervirulence determinants *iucABCD* and *rmpA2.* No plasmid structures were determined for these reference isolates by their uploaders, which made direct comparison impossible.

The closest neighbor of CriePt491 was the isolate 2566 (GCA_011044915, Saint Petersburg, Russia, 20 allele mismatches), which possessed the fusion plasmids mentioned above. This isolate had *bla_CTX-M-15_* and *bla_NDM-1_* beta-lactamase genes, which were also possessed by CriePt491. Interestingly, 2566 also possessed an apparently functional type I-E CRISPR-Cas system.

CgMLST profiles for the isolates studied and the corresponding reference isolates are presented in Appendix A.

## 4. Discussion

*K. pneumoniae* has already become a widespread healthcare-associated pathogen constituting a significant threat to public health due to its ability to rapidly acquire and spread AMR determinants and to develop resistance even to novel antibiotics [22,41]. Several lineages of “classical”, often MDR or XDR, *K. pneumoniae* strains, which have usually infected chronically ill patients residing in hospitals, were designated as high-risk clones due to their increased morbidity and mortality [3]. The best-characterized of these *K. pneumoniae* clones include, among others, ST147 and ST512 [3,42], to which the isolates described in our study belong. Although previously high-risk clones and hypervirulent clones of *K. pneumoniae* were considered to be separate evolutional groups, an increasing number of reports worldwide, especially in the last 5 years, have described the convergence of hypervirulence and MDR properties in single isolates of this species [17,19,43,44,45].

In this study, we analyzed three MDR ST512 isolates and one MDR ST147 isolate, which caused severe adverse conditions in patients admitted to a tertiary care hospital in which three out of the four patients died. Two ST512 isolates, CriePt492 and CriePt495, were almost identical both in terms of their genomic structure and in terms of their phenotypic characteristics, and the last ST512 isolate, CriePt494, was close to them in cgMLST profile but had a different plasmid repertoire. CriePt494 was the only isolate containing *peg-344*, *iucA* and the plasmid-borne *rmpA2* gene, which are considered reliable diagnostic markers for identifying strains in the hvKp-rich cohort [9,46], and it was assigned a virulence score of three on a five-point scale developed earlier [36]. All these genes, together with other virulence factors from the *iuc* cluster and several AMR genes, including *bla_NDM-1_*, were contained in a large 300k fusion conjugative plasmid, IncHI1B/IncFIB. A recent comprehensive in silico analysis revealed that about 70% of the known virulence plasmids from Genbank carried hybrid replicons of this type, and it specifically reported the genes *iucA* and *iutA*, which were found in the CriePt494_p2 plasmid, as being associated with high virulence [47]. The structure of this plasmid was similar to phvKpST147_NDM-1_2566 previously found in Russia [43] (the isolates with similar plasmids in that study belonged to ST15, ST147, ST395 and ST874), but their AMR gene content was different. Both CriePt492 and CriePt495 also had a conjugative plasmid IncHI1B/IncFIB each, but in this case, there were no AMR or virulence genes in these plasmids. The same was true for the IncHI1B/IncFIB plasmid of ST147 CriePt491, but it also lacked any relaxases and was predicted to be non-mobilizable. Thus, we can conclude that the possession of a fusion plasmid was not a property of the isolates having particular ST, and we reported its carriage in ST512, which had not been revealed previously by other researchers [43].

The formation of *K. pneumoniae* genetic lineages exhibiting both high virulence and carbapenem resistance can occur by several pathways associated with horizontal gene transfer [43,48], and the conjugation was considered its most significant mechanism [49]. Hypervirulent isolates can acquire MDR plasmids, and MDR isolates can also acquire plasmids carrying virulence factors. Some reports have supposed that the latter is more likely [21], but others have confirmed that the former occurs more often, or that both are possible [10,50], and the involvement of IncHI1B/FIB was revealed in both situations [51]. In our case, we can suppose the acquisition of additional virulence genes by the MDR isolate CriePt494 to form the fusion virulence/resistance-conferring plasmid since we revealed CriePt492 and 495, which (i) also contained IncHI1B/FIB, but without AMR or virulence factors, and (ii) included another IncFIB containing AMR, but not virulence, genes similar to the ones from IncHI1B/FIB of CriePt495, while the corresponding IncFIB from the latter did not contain any pathogenic factors. However, additional intermediate plasmids should be revealed to confirm the possible evolutionary pathway. Meanwhile, the formation of such mosaic plasmids by possible stepwise acquisition of transposon elements has been hypothesized in previous reports [43,52,53].

In general, the isolates studied showed great correspondence of their phenotypic and genomic resistance profiles. The resistance to gentamicin exhibited by CriePt491 and CriePt494 was highly likely conferred by the presence of the *armA* gene on the chromosome and the IncHI1B/FIB plasmid, respectively. Carbapenem resistance of the isolates could be attributed to NDM-1 and KPC-3 carbapenemases, as well as to mutations revealed in ompK36 and ompK37 porins, which were reported previously for carbapenem-resistant *K. pneumoniae* isolates from Turkey [54] also producing KPC-3, and from Nigeria [55]. Notably, CriePt491 had its *bla_NDM-1_* gene in a chromosome, which is quite unusual for *K. pneumoniae* and was previously associated with the integration from an IncHI1B-like plasmid [56]. All isolates exhibited a high level of AMR, and CriePt491 and CriePt494 were susceptible only to a ceftazidime/avibactam combination from the panel of the antibiotics tested.

CriePt491 was the only isolate not having conjugative plasmids and carrying a large fraction of its AMR genes on the chromosome, and these facts could be explained by its possession of a complete CRISPR-Cas type I-E system. The situation of chromosomal AMR gene location is not typical for *K. pneumoniae* since this species usually carries most of its acquired resistance genes on the plasmids [2,25,57], which was observed in other isolates from the current study. At the same time, all other isolates included the AcrIIA7 Cas9 inhibitor protein gene, which could be the reason why they did not have CRISPR-Cas, although AcrIIA7 has been supposed to target II-A systems [58]. The type I-E CRISPR system was previously suggested to efficiently limit the acquisition of antibiotic resistance genes and plasmids in *K. pneumoniae* strains [59,60,61]. Nevertheless, other recent findings have shown that the presence of a putative CRISPR-Cas system in a bacterial genome is not a limitation for the acquisition of common MDR plasmids by *K. pneumoniae* isolates [17,62], and some plasmids carrying resistance genes can even simultaneously contain a CRISPR-Cas system against other types of plasmids [53,63].

The targeting of CRISPR-Cas systems is achieved by incorporating the DNA fragments (protospacers) from infecting bacteriophages or plasmids to the CRISPR array, where they form a sequence-specific memory against subsequent invasions by the same bacteriophage or plasmid [63]. For example, I-E CRISPR-Cas-matched protospacers have commonly been found in conjugative IncFII-MDR plasmids [60], which play an important role in AMR dissemination within *K. pneumoniae* populations [64]. InxX3 plasmids also have a great conjugation ability and are more commonly associated with carbapenemase production than, e.g., IncR or IncH plasmids [65]. In our study, it was in fact the IncX3 plasmid that carried the *bla_KPC-3_* gene in ST512 isolates. The analysis of spacer sequences in the CriePt491 CRISPR-Cas system revealed a large number of spacers targeting IncFII and IncX3 plasmids, which this isolate did not possess. However, this purely correlative observation needs to be verified by additional experiments. Interestingly, a recent study has demonstrated the possibility of using the re-engineered conjugative endogenous CRISPR-Cas3 system to cure the AMR gene carrying *K. pneumoniae* IncFII plasmids in a *Galleria mellonella* infection model [64]. Thus, future investigations in this field seem to be very promising for tackling AMR in *K. pneumoniae* and other bacterial species.

The limitation of our study lies in the small number of the isolates investigated. However, the aim of this report was not to give a snapshot of *K. pneumoniae* spreading in a particular hospital or region, but rather to draw attention to the emergence of fusion plasmids carrying both virulence and resistance genes and to look for clues regarding the possible mechanisms of their formation. Prompt accumulation of such information is very important, and previous reports have revealed that timely and appropriate analysis of bacterial sequencing data could be a bottleneck in epidemiological surveillance [66,67]. We plan to continue the surveillance of MDR *K. pneumoniae* isolates and their plasmids in the future when more samples become available.

We believe that the plasmid data provided in this manuscript will facilitate the epidemiological surveillance of *K. pneumoniae* in Russia and other countries and that it presents useful data for developing prevention measures against this prominent pathogen. Such measures could possibly include the development of novel antibiotics [68] or other antimicrobial compounds like antimicrobial peptides [69,70], implementation of bacteriophages [71], and advancing antibiotic stewardship [72].

## 5. Conclusions

We have performed long- and short-read WGS and conducted comprehensive plasmid analysis of four clinical MDR *K. pneumoniae* isolates and we revealed a hybrid IncHI1B/FIB plasmid carrying both AMR and virulence genes in one of the isolates, CriePt494, belonging to a high-risk clone, ST512. This plasmid carried *peg-344*, *iucABCD* and *_p_rmpA2* high-virulence determinants, as well as multiple resistance genes, including *bla_NDM-1_* beta-lactamase. Other isolates also possessed plasmids of this type, but they did not carry any pathogenicity determinants, although their structure allows assuming that they are ‘ready’ to obtain AMR or virulence genes in the future. The fact that such plasmids were revealed in a rather small collection of clinical isolates suggests the possibility of their spread rate increasing and draws attention to the threat posed by the increasing risk of fusion AMR/virulence plasmid formation in clinical settings.

## Figures and Tables

**Figure 1 microorganisms-11-01314-f001:**
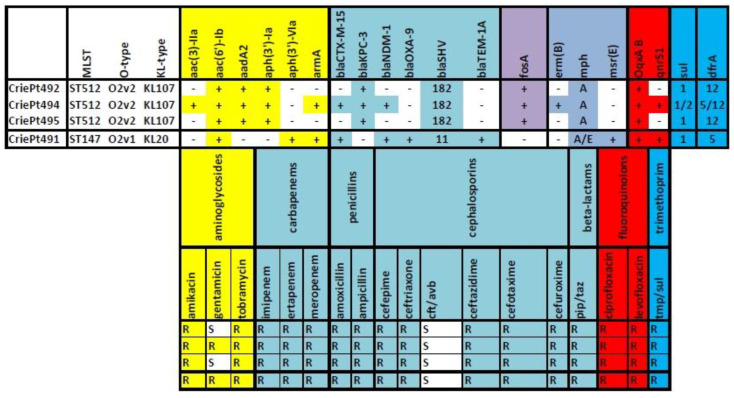
Phenotypic and genomic antibiotic resistance profiles for clinical *K. pneumoniae* isolates studied. Cft/avb—ceftazidime/avibactam; pip/taz—piperacillin/tazobactam; tmp/sul—trimethoprim/sulfamethoxazole.

**Figure 2 microorganisms-11-01314-f002:**
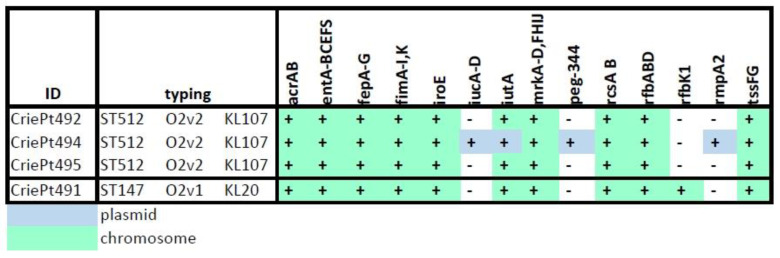
Important virulence factors revealed in the *K. pneumoniae* isolates studied.

**Figure 3 microorganisms-11-01314-f003:**
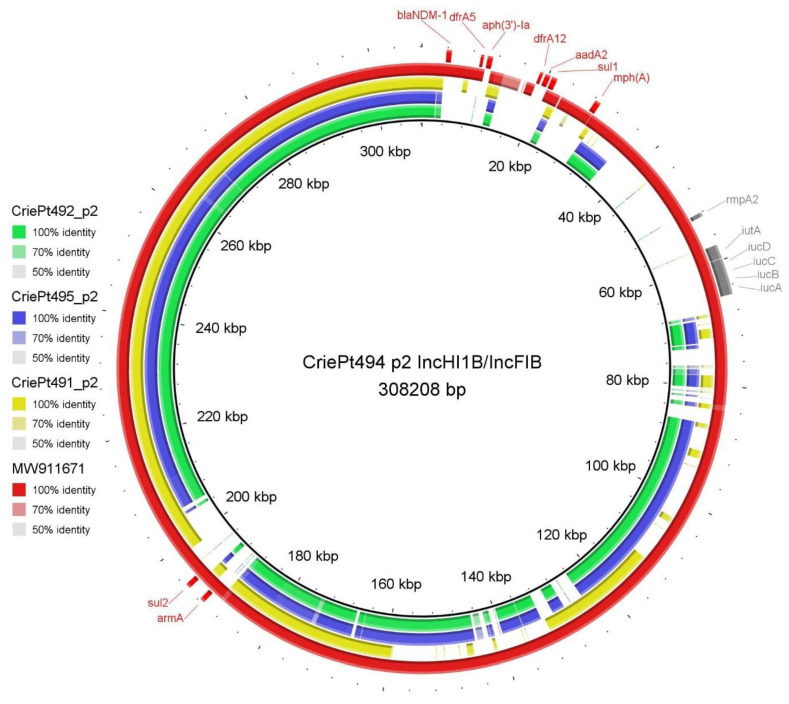
Circular diagram of IncHI1B/IncFIB plasmids from CriePt isolates and reference plasmid phvKpST147_NDM-1_2566. Antimicrobial resistance genes are indicated in red and virulence factors in gray.

**Figure 4 microorganisms-11-01314-f004:**
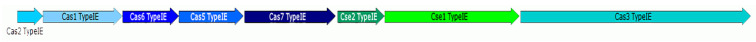
The scheme of the CRISPR-Cas system for *K. pneumoniae* isolate CriePt491.

**Figure 5 microorganisms-11-01314-f005:**
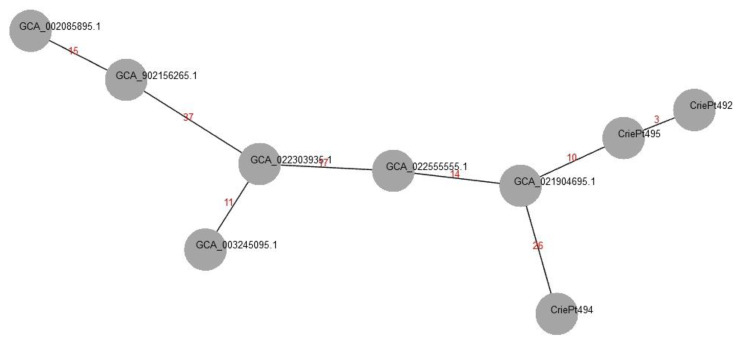
Minimum spanning cgMLST-based tree for ST512 *K. pneumoniae* isolates revealed by us and reference isolates from Genbank. Values of the pairwise allele differences between the isolates are given in red.

**Table 1 microorganisms-11-01314-t001:** Metadata for clinical *K. pneumoniae* isolates obtained from patients of ICU and pulmonology departments of a single tertiary care hospital.

Isolate Id	Patient Age (Years)	Patient Gender	Isolation Date	Department	Locus	Diagnosis (Main)
CriePt491	75	female	12 December 2021	ICU ^1^	BAL ^2^	Thrombosis of the superior mesenteric artery with necrosis of the colon
CriePt492	82	female	12 December 2021	ICU	BAL	Acute thrombosis of the femoral artery
CriePt494	64	male	8 December 2021	pulmonology	sputum	Interstitial pneumonitis
CriePt495	58	male	9 December 2021	ICU	BAL	Intracerebral parenchymal hemorrhage in the suprasellar region of the brain

^1^ ICU—intensive care unit; ^2^ BAL—bronchoalveolar lavage.

**Table 2 microorganisms-11-01314-t002:** Plasmid replicon composition for the clinical *K. pneumoniae* isolates studied.

Sample Id/Plasmids	num	Col(pHAD28)	ColRNAI	IncFIA	IncFIB	IncHI1B	IncR	IncX3	Unkn1 ^1^	Unkn2
CriePt491	5	+	-	-	+	+	+	-	-	+
CriePt492	6	+	+	-	+	+	-	+	+	-
CriePt494	7	+	+	+	+	+	-	+	+	-
CriePt495	6	+	+	-	+	+	-	+	+	-

^1^ Unkn1 and Unkn2 stand for two different types of non-typeable plasmids.

**Table 3 microorganisms-11-01314-t003:** Extended description of plasmids for the clinical *K. pneumoniae* isolates studied.

ID	Size, bp	GC Content	Replicon Type (s)	Relaxase Type	mpf_Type	Predicted Mobility	AMR/Virulence (VIR) Genes
CriePt491							
p2	179,707	0.448	IncHI1B/IncFIB	-	MPF_F	non-mobilizable	-
p3	109,649	0.493	IncFIB	-	-	non-mobilizable	-
p4	53,009	0.534	-	-	-	non-mobilizable	-
p5	46,984	0.534	IncR	-	-	mobilizable	AMR
p6	4915	0.431	Col(pHAD28)	-	-	non-mobilizable	-
CriePt492							
p2	250,630	0.461	IncHI1B/IncFIB	MOBH	MPF_F	conjugative	-
p3	184,725	0.533	IncFIB	MOBF	-	mobilizable	AMR
p4	53,292	0.496	IncX3	MOBP	MPF_T	conjugative	AMR
p5	10,689	0.475	Col(pHAD28)	-	-	non-mobilizable	AMR
p6	9730	0.532	ColRNAI	MOBC	-	mobilizable	-
p7	2963	0.650	-	MOBV	-	mobilizable	-
CriePt494							
p2	308,208	0.472	IncHI1B/IncFIB	MOBH	MPF_F	conjugative	AMR, VIR
p3	93,808	0.521	IncFIB	-	-	non-mobilizable	-
p4	89,753	0.510	IncFIA	MOBF	MPF_F	conjugative	AMR
p5	53,292	0.496	IncX3	MOBP	MPF_T	conjugative	AMR
p6	10,689	0.475	Col(pHAD28)	-	-	non-mobilizable	AMR
p7	9737	0.532	ColRNAI	MOBC	-	mobilizable	-
p8	2963	0.650	-	MOBV	-	mobilizable	-
CriePt495							
p2	251,830	0.461	IncHI1B/IncFIB	MOBH	MPF_F	conjugative	-
p3	190,476	0.533	IncFIB	MOBF	-	mobilizable	AMR
p4	53,292	0.496	IncX3	MOBP	MPF_T	conjugative	AMR
p5	10,689	0.475	Col(pHAD28)	-	-	non-mobilizable	AMR
p6	9730	0.532	ColRNAI	MOBC	-	mobilizable	-
p7	2963	0.650	-	MOBV	-	mobilizable	-

## Data Availability

The assembled genome sequences for all isolates were uploaded to the NCBI Genbank under the project number PRJNA942929.

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
