# Peer review of "Whole-Genome Sequencing Revealed the Fusion Plasmids Capable of Transmission and Acquisition of Both Antimicrobial Resistance and Hypervirulence Determinants in Multidrug-Resistant *Klebsiella pneumoniae* Isolates"

_microorganisms, 2023, doi:10.3390/microorganisms11051314_

Round 1

Reviewer 1 Report

In this study authors tried to described the detailed phenotypic and genomic characteristics of four MDR K. pneumoniae isolates based on antimicrobial susceptibility testing, hybrid whole-genome assembly, and phylogenetic comparison. This study is well design and has the potenial to be accepted and published after minor checks. 

Author Response

We thank the reviewer for reading our manuscript. We have checked the manuscript and correct minor errors we found.

Reviewer 2 Report

This is a genome-based study of a limited number of Klebsiella pneumonia isolates. In this regard, and although the work is scientifically correct, the originality and the significance of the content is low. In a situation where regular articles on molecular epidemiology analyze hundreds of isolates, the study of four strains (two of them, actually variants of the same) is not a particularly relevant contribution to the field.

To sum up, the work is OK, but the amount of novel information derived from this study does not justify its publication.

Author Response

This is a genome-based study of a limited number of Klebsiella pneumonia isolates. In this regard, and although the work is scientifically correct, the originality and the significance of the content is low. In a situation where regular articles on molecular epidemiology analyze hundreds of isolates, the study of four strains (two of them, actually variants of the same) is not a particularly relevant contribution to the field.

To sum up, the work is OK, but the amount of novel information derived from this study does not justify its publication.

We thank the reviewer for reading our manuscript, but we cannot agree with the conclusions given. First, the ‘regular articles’ in molecular epidemiology can in fact analyze hundreds of the isolates, but the manuscripts presenting whole genome data for such number of the isolates, especially using long-read sequencing, are relatively rare.

 Second, the quantity does not always assure quality, namely, the contribution can be estimated by the conclusions and novel information derived from the analyzed data, and not from the number of the genomes studied itself. Currently, more than 10000 partially or fully sequenced Klebsiella pneumoniae genomes are available in Genbank, but this does not mean that this bacterium is completely known to the researchers and no further data can be obtained. In fact, the contrary is usually true since the mechanisms of resistance and virulence acquisition for K. pneumoniae are still far from being clearly understood despite such a large number of genomes being available.

 Third, our manuscript, as well as the Special Issue itself, is not dedicated to studying the global epidemiology of K. pneumoniae in a particular region or hospital, but rather to providing insights into novel plasmid formation and the mechanisms of virulence and resistance gene acquisition for a limited set of the isolates. Although two isolates from our study were in fact close in their chromosome sequences, the slight differences in their plasmid composition provided interesting insights into mobile element evolution.

The novelty of the manuscript can be described as follows:

  • We obtained the precise hybrid AMR/virulence plasmid structure, as well as the structures of other AMR carrying plasmids, using long-read sequencing, and revealed significant differences with the reference plasmids previously published
  • For the first time, we revealed hybrid IncHI1B/FIB plasmid carrying both AMR and virulence genes in ST512 isolates from Russia
  • We have revealed other isolates possessing the plasmids of this type, but they did not carry any pathogenicity determinants, although their structure allows assuming that they are ‘ready’ to obtain AMR or virulence genes in future, which provides useful insights into plasmid evolution and mechanisms of AMR acquisition for K. pneumoniae
  • We have shown that a functional CRISPR-Cas system can prevent the acquisition of conjugative plasmids in pneumoniae

In addition, the isolates described in the manuscript were just a representative subset of all K. pneumoniae isolates studied, and we added this information to our manuscript (Section 2.1).

Reviewer 3 Report

·         Line 18: The “the” before “horizontal” and “plasmid” should be deleted as they seem redundant in order to make the sentence more comprehensible.

·          Lines 50-51: Consider using the earlier defined abbreviation “HvKp” or the phrase “The pathogen” in place of “Hypervirulent K. pneumoniae” to improve the readability and clarity of the sentence.

·         Line 69: The “Namely” preceding the sentence is not necessary and should be removed.

·         Liine 93: What do the authors mean by “during the close period (3-4 days) of their stay”? To eliminate any ambiguity, the statement must be elucidated.

·         Line 94: Consider replacing the word “the” with “a” and “of” with “in the” to make the sentence read better.

·         Line 97: The title of the Metadata table must be rephrased to reflect the source of the K. pneumoniae isolates studied. In its current state and considering the list of the “Isolate Ids” it seems to present that for example, Isolate CriePt491 is female aged 75. The units of “Age” in the column must be defined. E.g.: Age(years). Also, the content of the “Diagnosis (main)” column does not really correspond with the other components of the table especially the last row

·         Line 125-130: The idea to use both long and short-read sequencing platforms coupled with the Hybrid short- and long-read assembly methodologies sounds great. However, were there any significant differences noted in the results of these assembly strategies if considered independently?

·         Line 203: Authors must show a summary of the Point mutation analysis perhaps as a piece of supplementary information to allow for a better appreciation of the presented statement.

·          Line 244:  There is an interchange of “CriePt…” and “CriePir…” in the isolate ID annotation. This is also seen in Table 3. Authors must stick to one naming format unless the two connote different things. If so, they must be defined.

·         Line 254: Consider inserting “together” between “discussed” and “the” to bring clarity to the discussion.

·         Line 409-411: For improved readability, the sentence must read “In this study, we analyzed three MDR ST512 isolates and one MDR ST147 isolate, which caused severe adverse conditions in patients admitted to a tertiary care hospital in which death three out of the four patients died.”

Author Response

We would like to thank the reviewer for useful suggestions that has led to significant improvements in our manuscript content and readability. Our comments are given below boldfaced.

  • Line 18: The “the” before “horizontal” and “plasmid” should be deleted as they seem redundant in order to make the sentence more comprehensible.

Fixed as suggested

  • Lines 50-51: Consider using the earlier defined abbreviation “HvKp” or the phrase “The pathogen” in place of “Hypervirulent K. pneumoniae” to improve the readability and clarity of the sentence.

Fixed as suggested

  • Line 69: The “Namely” preceding the sentence is not necessary and should be removed.

Fixed as suggested

  • Liine 93: What do the authors mean by “during the close period (3-4 days) of their stay”? To eliminate any ambiguity, the statement must be elucidated.

We meant that the samples were collected within 3-4 days from each other. We clarified this as ‘isolates were collected… during short interval (3-4 days) within their hospital stay’.

  • Line 94: Consider replacing the word “the” with “a” and “of” with “in the” to make the sentence read better.

Fixed as suggested

  • Line 97: The title of the Metadata table must be rephrased to reflect the source of the K. pneumoniae isolates studied. In its current state and considering the list of the “Isolate Ids” it seems to present that for example, Isolate CriePt491 is female aged 75. The units of “Age” in the column must be defined. E.g.: Age(years). Also, the content of the “Diagnosis (main)” column does not really correspond with the other components of the table especially the last row

Fixed as suggested

  • Line 125-130: The idea to use both long and short-read sequencing platforms coupled with the Hybrid short- and long-read assembly methodologies sounds great. However, were there any significant differences noted in the results of these assembly strategies if considered independently?

The main advantage of hybrid assembly is that it allows obtaining the correct plasmid structures.  Plasmid sequences usually contain a number of repeats, which represents a hindrance for short-read assembler software, so that the plasmid could not be assembled down to a single contig or, which is even worse, some similar repetitive parts of different plasmids could be assembled into a single sequence. In contrast, long reads allow reading the whole plasmid, or at least its part up to 100k base pairs, which greatly facilitates the assembly process. As to the long-read assembly alone, it was previously shown to be prone to systematic reading errors, especially in homopolymeric sequences, and short reads can be used to ‘polish’ them in order to remove such errors from assembly. These advantages of hybrid assembly were also previously reported and confirmed by other researchers (e.g., 10.3390/microorganisms9122560,  https://doi.org/10.1186/s12864-020-07041-8, https://doi.org/10.1016/j.envpol.2021.117856).

  • Line 203: Authors must show a summary of the Point mutation analysis perhaps as a piece of supplementary information to allow for a better appreciation of the presented statement.

The summary of point mutation analysis was

  • Line 244:  There is an interchange of “CriePt…” and “CriePir…” in the isolate ID annotation. This is also seen in Table 3. Authors must stick to one naming format unless the two connote different things. If so, they must be defined.

Thank you for pointing out this issue. The correct names are ‘CriePt’. We fixed this in the text and tables.

  • Line 254: Consider inserting “together” between “discussed” and “the” to bring clarity to the discussion.

Fixed as suggested                 

  • Line 409-411: For improved readability, the sentence must read “In this study, we analyzed three MDR ST512 isolates and one MDR ST147 isolate, which caused severe adverse conditions in patients admitted to a tertiary care hospital in which death three out of the four patients died.”

Fixed as suggested

Round 2

Reviewer 2 Report

I appreciate the detailed answers of the authors. However, I still think that the degree of novelty is low. Nevertheless, it is true, and this is stated in my ranks, that science is solid, supports the conclusions and the quality of the presentation is OK. Hence if novelty is not a major issue for the editor and the other referees, I have nothing against the publication of the work.